# Understanding the mechanism of red light-induced melatonin biosynthesis facilitates the engineering of melatonin-enriched tomatoes

**Zixin Zhang** [1], **Xin Zhang**[1], **Yuting Chen**[1], **Wenqian Jiang**[1], **Jing Zhang**[1], **Jiayu Wang**[1], **Yanjun Wu**[1], **Shouchuang Wang** [2], **Xiao Yang**[3], **Mingchun Liu** [1] & **Yang Zhang** [1] ✉

Melatonin is a functionally conserved broad-spectrum physiological regulator found in most biological organisms in nature. Enrichment of tomato fruit with melatonin not only enhances its agronomic traits but also provides extra health benefits. In this study, we elucidate the full melatonin biosynthesis pathway in tomato fruit by identifying biosynthesis-related genes that encode caffeic acid O-methyltransferase 2 (SlCOMT2) and N-acetyl-5-hydroxytryptamine-methyltransferases 5/7 (SlASMT5/7). We further reveal that red light supplementation significantly enhances the melatonin content in tomato fruit. This induction relies on the "serotonin−N-acetylserotonin−melatonin" biosynthesis route via the SlphyB2-SlPIF4-SlCOMT2 module. Based on the regulatory mechanism, we design a gene-editing strategy to target the binding motif of SlPIF4 in the promoter of *SlCOMT2*, which significantly enhances the production of melatonin in tomato fruit. Our study provides a good example of how the understanding of plant metabolic pathways responding to environmental factors can guide the engineering of health-promoting foods.

Melatonin (N-acetyl-5-methoxytryptamine) is an indoleamine compound found in all organisms from plants to animals. It was first discovered in the pineal gland of cattle in 1958 and is also known as epiphysin[1,2], and it has been shown to be the most powerful endogenous free radical scavenger known at present[3,4]. In animals and humans, melatonin has the functions of improving sleep, delaying aging, alleviating allergic symptoms, and regulating the immune system[5,6]. Some studies have also shown the oncostatic property of melatonin on different types of tumors, as well as reducing the damage resulting from inflammation[7,8].

In plants, melatonin mainly functions as a growth promoter and antioxidant[9]. It has the activities of delaying senescence, enhancing photosynthesis, regulating the photoperiod, affecting seed germination and root morphogenesis, regulating flowering and fruit ripening, removing free radicals, and alleviating stress damage, and it can give plants the ability to resist adverse environments, which is conducive to plant survival and reproduction[9,10].

As the world's favorite fruit, tomato is the ideal target for plant metabolic engineering[11]. Synthetic strategies have been successfully applied to tomato metabolic engineering. Fruit-specific expression of the transcription factors *AmDel* and *AmRos 1* leads to the upregulation of genes required for anthocyanin biosynthesis and results in increased anthocyanin levels and higher total antioxidant capacity[12,13]. The fruit-specific expression of *AtMYB12* could be used to enhance the

[1]Key Laboratory of Bio-resource and Eco-environment of Ministry of Education, College of Life Sciences, Sichuan University, Chengdu 610065, China. [2]Sanya Nanfan Research Institute of Hainan University, Hainan Yazhou Bay Seed Laboratory, Sanya 572025, China. [3]Institute of Urban Agriculture, Chinese Academy of Agricultural Sciences, Chengdu National Agricultural Science & Technology Center, Chengdu 610213, China. ✉e-mail: yang.zhang@scu.edu.cn

demand for aromatic amino acid biosynthesis, and it can be applied as an effective tool to engineer palpable levels of phenylpropanoids in tomato[14,15]. During the past several years, the rapid development of genome-editing technology has provided useful tools for creating good tomato germplasm. The accumulation of provitamin D3 in tomatoes was engineered by genome editing, which provides a bio-fortified food with the added possibility of supplemental production from waste material[16]. By inducing mutations at the C-terminal region of *GAD* genes utilizing the CRISPR/Cas9 system, the content of γ-aminobutyric acid (GABA) was greatly increased in tomato leaves and red-stage fruits[17]. In 2021, the world's first GABA-enhanced genome-edited tomato, 'Sicilian Rouge', made with CRISPR–Cas9 technology was launched into the open market[18].

Previously, tomato fruit treated with exogenous melatonin was found to show higher levels of nutrients (such as carotenoids, flavor, etc.) with better fruit yields compared to nontreated plants[19,20]. Moreover, melatonin treatments effectively promote fruit ripening while maintaining the sensory and nutritional attributes of fruit by enhancing antioxidant capacity in ripening fruit, which refers to delaying fruit senescence and extending shelf life[19–21]. Therefore, increasing the content of melatonin in tomato fruit may improve both nutrition and agronomic traits.

Previous studies have shown that the synthesis of melatonin in plants starts from the synthesis of tryptophan, which requires four consecutive enzymatic reactions. Tryptophan decarboxylase (TDC) and tryptophan-5-hydroxylase (T5H) are key enzymes in the first two steps of melatonin synthesis, catalyzing the production of serotonin (5-hydroxytryptamine), 5-hydroxytryptamine-N-acetyltransferase (SNAT) and n-acetyl-5-hydroxytryptamine-methyltransferase (ASMT)/ caffeic acid-o-methyltransferase (COMT), catalyzing the final forma-tion of melatonin from serotonin. Studies have found that there are at least four possible melatonin synthesis routes in plants, and TDC and SNAT may be the rate-limiting enzymes in the process of melatonin synthesis[22,23]. However, it has also been suggested that ASMT may be the rate-limiting enzyme in the process of melatonin synthesis[24–27]. On the other hand, COMT can effectively catalyse the production of melatonin, showing strong ASMT activity. A previous study showed that melatonin contents were significantly reduced in *Arabidopsis comt* knockout mutants[28].

To date, it has been found that there are at least five *TDC* candidate genes in tomato, of which *SlTDC3* is expressed in almost all tissues, and *SlTDC1* and *SlTDC2* are only expressed in tomato fruits and leaves, respectively, indicating that the expression of *TDC* genes may be tissue-specific, and the expression of these genes may play different roles in plant growth and development or resistance[29,30]. However, the full melatonin biosynthesis pathway, especially in tomato fruit, has yet to be elucidated.

The synthesis and signal transmission of melatonin in plants is significantly affected by environmental factors (such as light and temperature)[31,32]. The regulation of melatonin synthesis by light signals has been well-studied in animals. Studies in mice have shown that melatonin synthesis depends on the rhythm clock and the core reg-ulator cry1/2 of the light-sensing signal[33]. However, research on plants is lagging behind. How different light signals coordinate the synthesis and metabolism of melatonin has been unclear.

Here, the melatonin synthesis pathway in tomato fruit was com-pletely elucidated. The functions of biosynthesis-related genes (*SlSNAT, SlASMT5, SlASMT7,* and *SlCOMT2*) were identified. We also found that red light treatment significantly promoted melatonin synthesis in tomato fruit via the SlphyB2-SlPIF4-SlCOMT2 module. Based on the regulatory mechanism, we targeted the binding motif of PIF4 in the promoter of *SlCOMT2* to design a gene-editing strategy and significantly enhanced the production of melatonin in tomato fruit. Our data not only expand our current knowledge of how environ-mental factors affect the biosynthesis of key metabolites but also

provide a good example of how to use the regulatory mechanism to guide the breeding of crops with enhanced nutrition.

## Results

### Screening of melatonin biosynthesis-related genes in tomato fruit

To elucidate the full melatonin biosynthesis pathway in tomato, we performed a BLAST search of the tomato genome for genes homo-logous to known melatonin biosynthesis-related genes: *TDC, T5H, SNAT, ASMT,* and *COMT*. In total, 17 candidates were identified (Fig. S1a). During the tomato fruit development process, the content of melatonin increases significantly at the breaker stage (Fig. 1a). Using the transcriptome data from the MicroTom Metabolic Network (MMN)[34], we further conducted correlation (Figs. S1b, 1b) and quanti-tative (Fig. S1c) analyses to narrow down 10 candidate genes with reasonable expression levels in tomato fruit (Fig. 1b).

All 10 candidate genes were then verified by transient over-expression in tobacco (*Nicotiana Benthamiana*) leaves and silencing in tomato fruits. We found that *Solyc07g054280*, which encodes SlTDC2, is responsible for the first step from tryptophan (1) to tryptamine (2) in melatonin biosynthesis. *Solyc09g014900*, which encodes SlT5H, cata-lyses the next step from tryptamine (2) to serotonin (3) (Fig. 1c, Fig. S2, S3, S4, S5, and S6).

Previous studies indicate that from serotonin (**3**), melatonin (**6**) biosynthesis might have alternative routes, which require the partici-pation of SNAT, ASMT, or COMT[9,22,23]. We transiently overexpressed/ silenced the remaining 7 genes to check the contents of melatonin. Transient overexpression of *SlASMT5* (Solyc03g097700), *SlASMT7* (Solyc06g064500), *SlCOMT2* (Solyc10g85830), and *SlSNAT* (Solyc10g074910) can significantly induce the production of melato-nin in tobacco leaves. Silencing of these genes significantly reduced the melatonin content in tomato fruit (Fig. 1c, Figs. S2, S3, S4, S5, and S6). Notably, the expression level of *SlASMT7* was found to be asso-ciated with the 5-methoxytryptamine (**4**) route of melatonin bio-synthesis (Fig. 1c). All these data indicate that SlSNAT, SlCOMT2, SlASMT5, and SlASMT7 are involved in the biosynthesis of melatonin.

### Functional verification of the roles of SlSNAT, SlCOMT2, and SlASMT5/7 in melatonin biosynthesis

The expression levels of *SlASMT5, SlASMT7, SlCOMT2,* and *SlSNAT* in different tissues were measured quantitatively. While *SlSNAT* was expressed in all tested tissues, *SlASMT7, SlASMT5,* and *SlCOMT2* were mainly expressed in fruits after the breaker stage (Fig. S7). The expression levels of these genes matched previous transcriptome data (Fig. S1a). Figure S8 shows the expression of these four genes in the Tomato Expression Atlas database[35] and MMN database[34]. This is consistent with the melatonin content (Fig. 1a). The localization experiment using *Arabidopsis* protoplasts showed that SlSNAT was localized in the chloroplast, while SlASMT5, SlASMT7, and SlCOMT2 were in the nucleus and cytoplasm (Fig. S9). This is consistent with previous reports[22,23,30].

We then generated stable overexpression and RNAi lines for *SlASMT5, SlASMT7, SlCOMT2,* and *SlSNAT* (Figs. S10, S11). Compared to WT fruit, overexpression of these four genes individually significantly enhanced the production of melatonin, while silencing of these genes reduced the content of melatonin (Fig. 2a, Fig. S12).

In vitro enzyme assays using recombinant proteins from *E. coli* confirmed that SlSNAT can catalyse the production of N-acetylserotonin (**5**) from serotonin (**3**), as well as the synthesis of melatonin from 5-methoxytryptamine (**4**) (Fig. 2b and Fig. S13). This indicates that the biosynthesis of melatonin in tomato fruit may have two possible routes: one is through the "serotonin(**3**)−N-acetylser-otonin (**5**)−melatonin (**6**)" route, and the other is the "serotonin (**3**)−5-methoxytryptamine (**4**)−melatonin (**6**)" route. We then incubated the recombinant SlASMT5, SlASMT7, and SlCOMT2 proteins with either

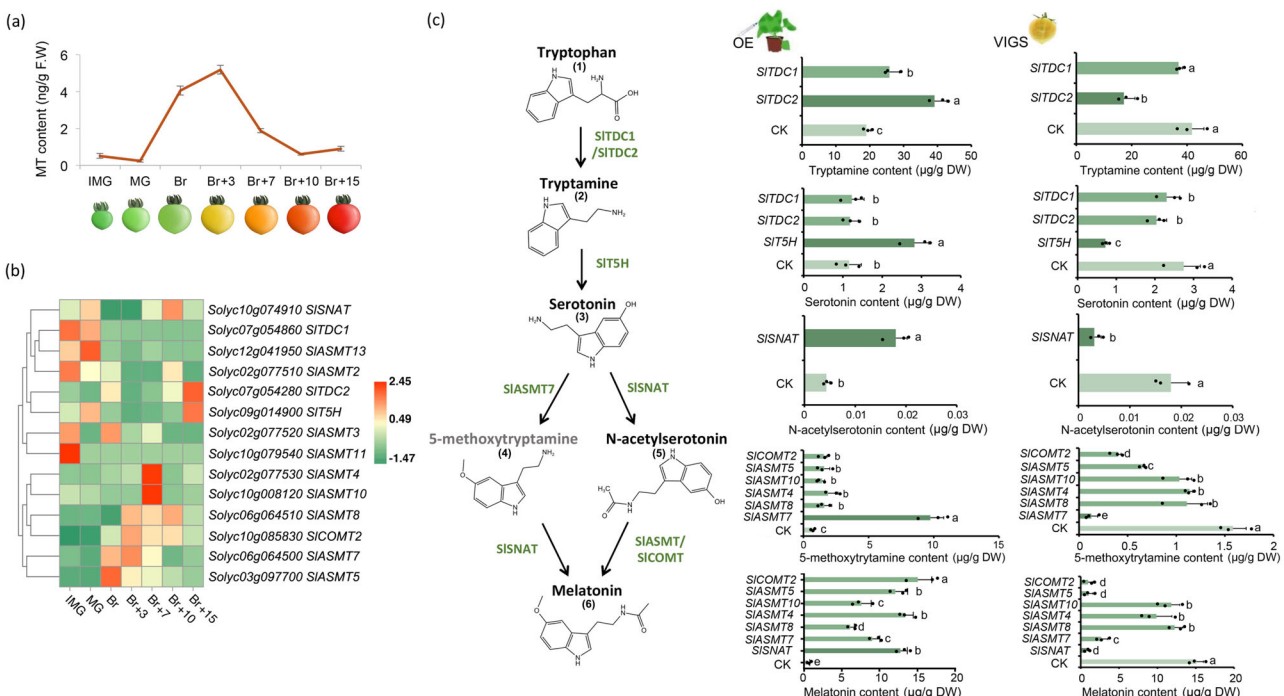

**Fig. 1 | Preliminary screening of key structural genes for melatonin biosynthesis in tomato fruit. a** Determination of melatonin content in tomato fruit at different development stages. IMG (immature green), MG (mature green), Br (breaker), Br+n (breaker plus n days). **b** The expression analysis heat map of the melatonin biosynthetic genes was obtained by screening from the MMN Database. **c** Melatonin biosynthetic gene obtained by instantaneous verification screening. 'OE' indicates gene transient overexpression (injection into tobacco leaves); 'VIGS' indicates gene transient silencing (injection into tomato fruits). CK indicates the determination result after injection of the corresponding empty carrier. 10–12 individual leaves (for transient expression) or 10–12 individual tomato fruits (for VIGS) with uniform sizes were pooled as one biological replicate. Data are represented as Mean ± SEM ($n = 3$). The $P$ values indicate the results from pairwise comparisons of one-way ANOVA tests. Different letters represent a significant difference at $P < 0.05$.

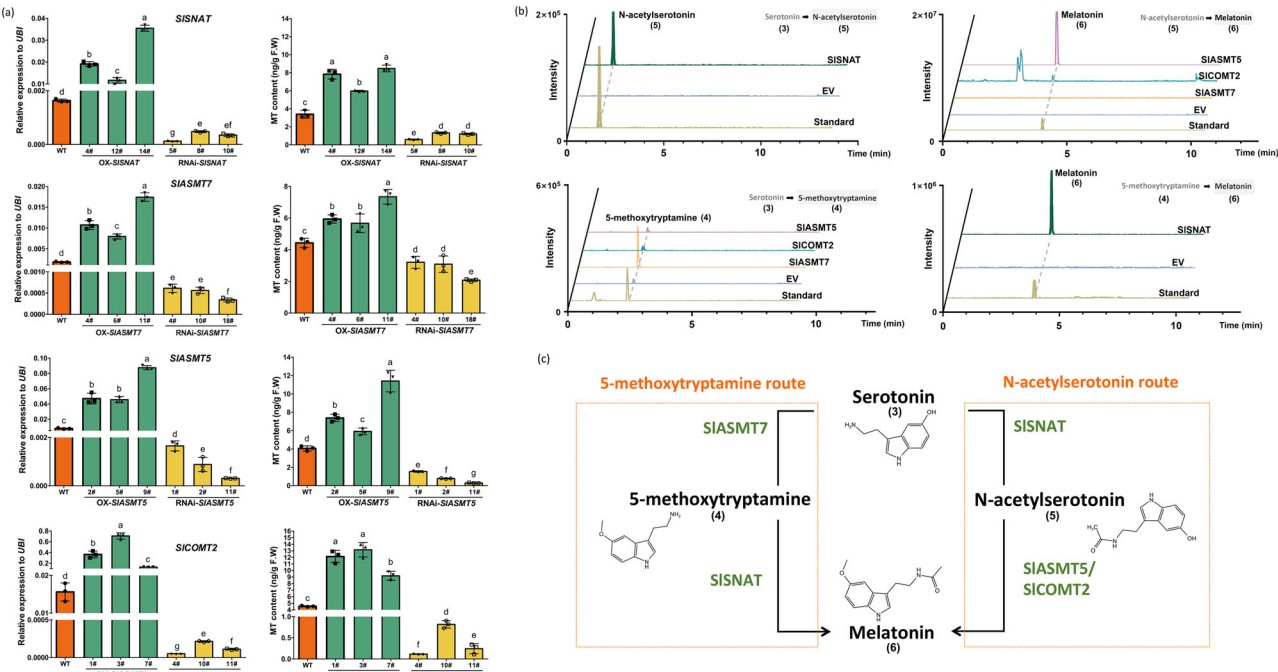

**Fig. 2 | In vivo and in vitro verification of melatonin biosynthetic genes.**
**a** Determination of gene expression and melatonin content in stable transgenic tomato. OX-*SlGENE* represents the overexpression lines, and RNAi-*SlGENE* represents the silencing lines. 10–12 individual tomato fruits at the Br+3 stage were pooled as one biological replicate. Data are represented as Mean ± SEM ($n = 3$). The $P$ values indicate the results from pairwise comparisons of one-way ANOVA tests. Different letters represent a significant difference at $P < 0.05$. Source data are provided as a Source Data file. **b** In vitro enzyme activity verification of key structural genes. Different proteins were incubated with different substrates (serotonin (**3**), N-acetylserotonin (**5**) and 5-methoxytryptamine (**4**), respectively) to detect the production of N-acetylserotonin (**5**), 5-methoxytryptamine (**4**), and melatonin (**6**), respectively. 'EV' indicates the empty vector for negative control. **c** The roles of SlSNAT, SlCOMT2, and SlASMT5/7 in melatonin biosynthesis.

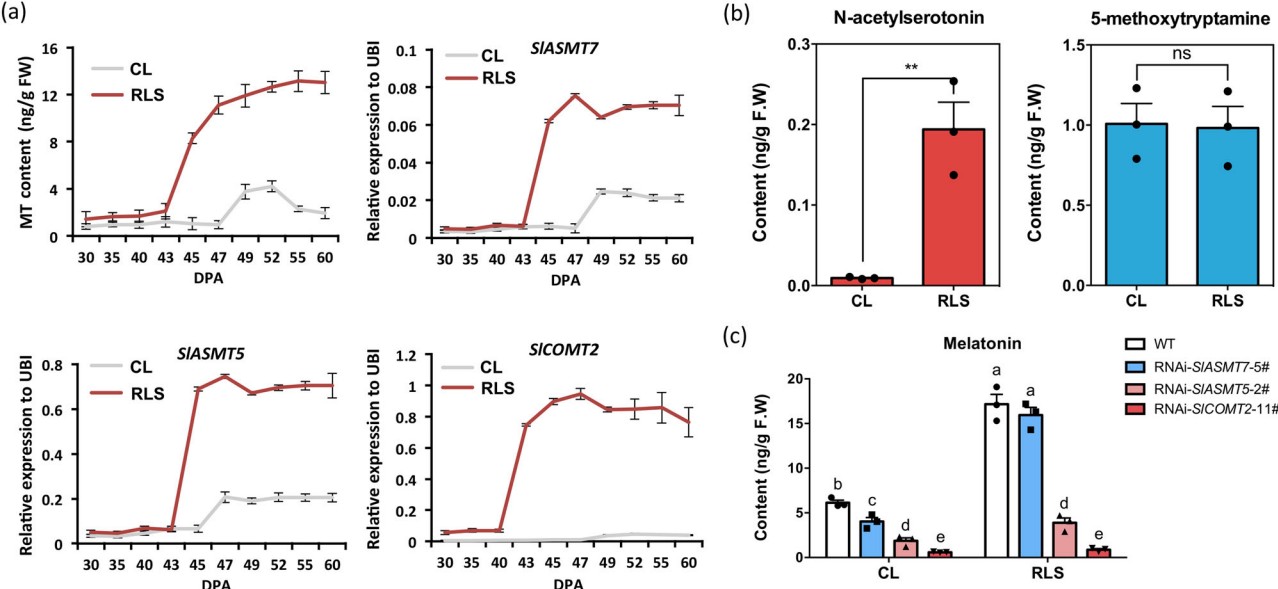

**Fig. 3 | Red light supplement-induced melatonin biosynthesis in tomato fruit.**
**a** Red light can significantly induce melatonin synthesis in tomato fruit. Determination of melatonin content and quantification of biosynthetic genes in tomato fruit at 10 different development stages under control light (CL) and red light supplement (RLS). Data are represented as mean ± SD ($n = 3$). In which 10–12 individual fruits were pooled as one biological replicate. **b** Content of N-acetylserotonin (**5**) and 5-methoxytryptamine (**4**) in wild-type tomato fruit after red light treatment. Data are represented as Mean ± SD ($n = 3$). In which 10–12 individual fruit were pooled as one biological replicate. * indicates a significant difference from control light (CL) analyzed by two-sided Student's $t$ test. (**$P = 0.0054$), ns ($P = 0.8949$), ns indicates not significant ($P > 0.05$). Source data are provided as a Source Data file. **c** Melatonin content in fruit of transgenic tomato lines after red light treatment. Data are represented as Mean ± SEM ($n = 3$). In which 10–12 individual fruits were pooled as one biological replicate. The $P$ values indicate the results from pairwise comparisons of one-way ANOVA tests. Different letters represent a significant difference at $P < 0.05$. Source data are provided as a Source Data file.

serotonin or N-acetylserotonin. The recombinant SlASMT7 can catalyse the formation of 5-methoxytryptamine from serotonin but failed to produce melatonin from N-acetylserotonin (Fig. 2b and Fig. S13). This indicates that SlASMT7 is involved in the 5-methoxytryptamine route of MT biosynthesis. On the other hand, both recombinant SlASMT5 and SlCOMT2 can catalyse the production of melatonin from N-acetylserotonin while failing to produce 5-methoxytryptamine from serotonin (Fig. 2b and Fig. S13). We further verified the function of SlASMT5, SlASMT7, and SlCOMT2 in vivo by RNAi and found that only silencing SlASMT7 significantly reduced the contents of 5-methoxytryptamine (Fig. S14). These results indicate that SlASMT5 and SlCOMT2 catalyse the N-acetylserotonin route of MT biosynthesis, while SlASMT7 is involved in the 5-methoxytryptamine route (Fig. 2c).

### Melatonin biosynthesis in tomato fruit is significantly induced by red light treatment
Red light supplementation was reported to introduce excellent characterisics such as early ripening, enhanced nutrients, and delayed senescence to tomato fruit[36–39]. We found that when tomato plants were provided with red light supplementation, the expression of SlASMT7, SlASMT5, and SlCOMT2 was induced, and the content of melatonin was significantly increased during the ripening process (Fig. 3a, S15). We also found that red light supplementation significantly increased the content of N-acetylserotonin but not 5-methoxytryptamine (Fig. 3b, S16). It seems that red light-induced melatonin biosynthesis relies on the activation of the N-acetylserotonin route via SlCOMT2 and SlASMT5.

To test this hypothesis, we repeated the red light supplementation experiment for the SlASMT7, SlASMT5, and SlCOMT2 RNAi lines. Compared to WT, the fruit of RNAi-SlASMT5 and RNAi-SlCOMT2 lines contained significantly lower melatonin under red light supplementation, while the RNAi-SlASMT7 line still responded well to red light treatment (Fig. 3c). All these data indicate that red light

supplementation enhances melatonin biosynthesis in tomato fruit via activation of the expression levels of SlASMT5 and SlCOMT2.

### SlPIF4 directly inhibits the expression of SlCOMT2 to suppress melatonin biosynthesis in tomato fruit
To investigate the molecular mechanism of red light-induced melatonin biosynthesis, we scanned the promoter regions of SlASMT5 and SlCOMT2. A series of light signal-related G-box elements were found in both promoters (Figs. 4a, S17a). On the other hand, proSlASMT5 and proSlCOMT2 were used as baits to screen yeast one-hybrid libraries. A cDNA fragment showing homology to phytochrome-interacting factors 4 (SlPIF4) was identified to bind to both proSlCOMT2 and proSlASMT5 (Supplementary Data 1). PIF4 has been reported to play vital roles in the light response and is capable of binding to the G-box domain[40]. Therefore, we hypothesize that SlPIF4 is a potential regulator of MT biosynthesis.

To investigate whether SlPIF4 can directly bind to proSlCOMT2 and proSlASMT5, we first performed a yeast one-hybrid (Y1H) assay. Three G-box elements of the SlCOMT2 genome sequence were selected as possible binding sites (P1-P3) (Fig. 4a). The results showed that SlPIF4 could bind to the P2 element of the SlCOMT2 promoter (Fig. 4b). A G-Box was also predicted on the promoter of SlASMT5, but it could not be bound by SlPIF4 (Fig. S17).

Using the Dual-Luc system in both tobacco leaves (Fig. 4c) and tomato protoplasts (Fig. 4d), we further confirmed that SlPIF4 could repress the activity of the SlCOMT2 promoter. When the P2 motif was mutated, the inhibition of proCOMT2 by SlPIF4 was released (Fig. 4f). EMSA with normal and mutated probes with the CArG motif in the promoter (P2 and mP2) of proSlCOMT2 also suggested that SlPIF4 directly binds to the SlCOMT2 promoter (Fig. 4e).

To further examine the direct binding of SlPIF4 to proSlCOMT2 in vivo, we generated FLAG-tagged SlPIF4-overexpressing tomato lines (Figs. S18, S19). By ChIP–qPCR, we found that SlPIF4 directly binds to

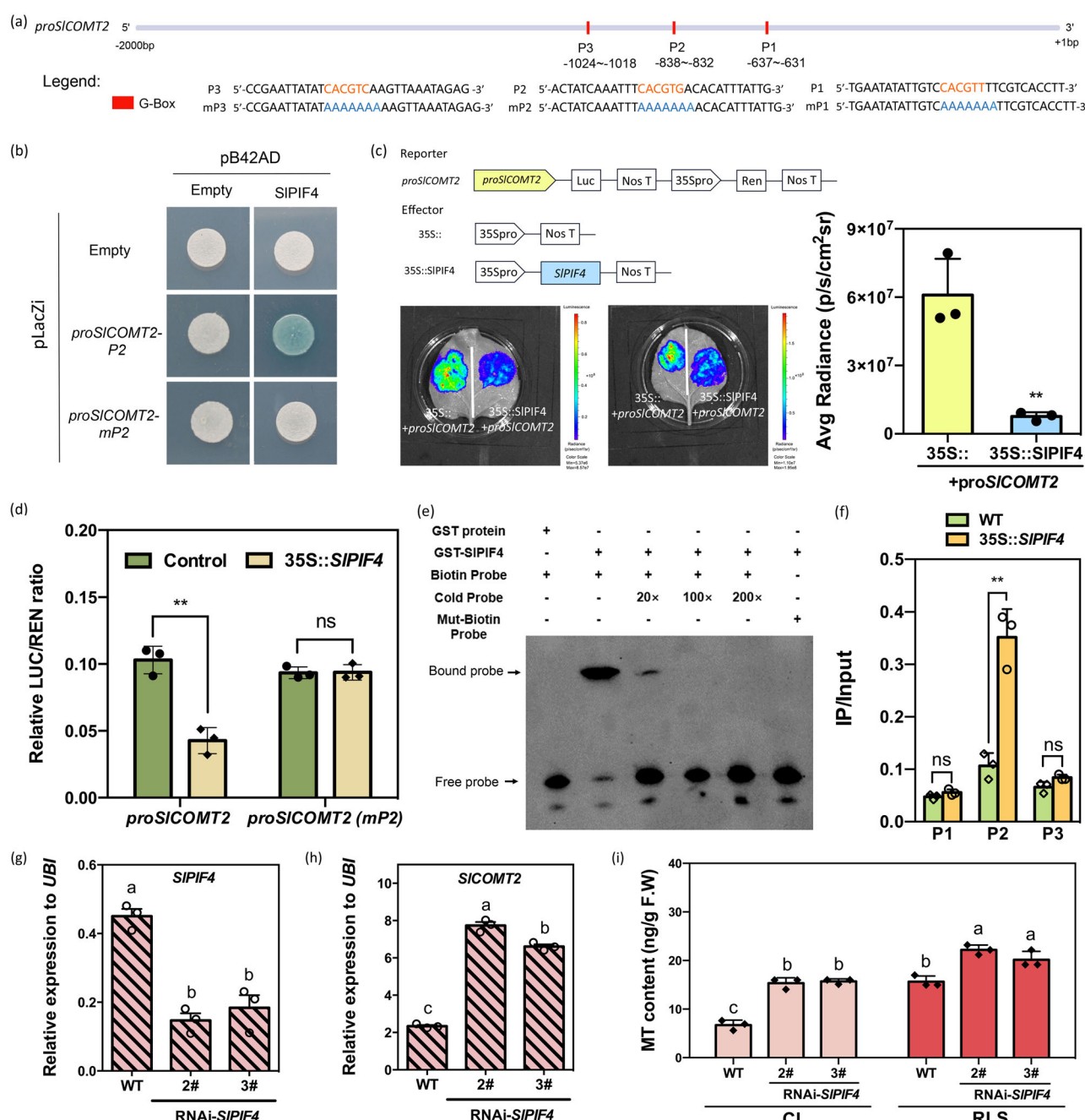

**Fig. 4 | SlPIF4 directly binds to the G-Box domain (P2) of *proSlCOMT2* to inhibit its expression. a** Schematic diagrams showing the SlASMT5 and SlCOMT2 genomic regions. The position of G-BOX is indicated by a red BOX. **b** Interactions between SlPIF4 proteins and *SlCOMT2* promoters with P2 and P2 mutation (mP2) in yeast cells. A blue plaque indicates binding. **c** Interactions of SlPIF4 protein and the promoters of *SlCOMT2* confirmed with dual-luciferase reporter assays in *Nicotiana benthamiana* leaves. 35 S::+*proSlCOMT2* were used as controls. The right column chart shows the quantitative fluorescence intensity. (**$P = 0.0045$). * indicates significant difference from 35 S:: empty vector analyzed by two-sided Student's *t* test. **d** SlPIF4 binding to the regions of *proSlCOMT2* in the wild-type (WT) and transgenic lines of 35 S::*SlPIF4*. '*proSlCOMT2(mP2)*' is a 2000 bp promoter sequence with a mutation in the P2 domain. LUC/REN is the average ratio of the bioluminescence of firefly luciferase to that of Renilla luciferase. (**$P = 0.0018$), ns ($P = 0.9387$), ns ($P = 0.0505$, P2), ns indicates not significant ($P > 0.05$). * indicates significant difference from control analyzed by two-sided Student's *t* test. **e** EMSA of SlPIF4 binding to the P2/mP2 fragment. SlPIF4 binds to the P2 fragment of *proSlCOMT2*, while the mutant of P2 (mP2) does not present binding. '+' indicates presence; and '-' indicates absence. **f** ChIP analysis of SlPIF4 binding to the regions of SlCOMT2 in the WT and transgenic lines of 35 S::*SlPIF4*. (**$P = 0.0020$), ns ($P = 0.1222$, P1), ns ($P = 0.0505$, P2), ns indicates not significant ($P > 0.05$). * indicates significant difference from WT analyzed by two-sided Student's *t* test. This experiment was repeated independently two times with similar results. Data in **c, d, f** are represented as Mean ± SD ($n = 3$), Student's *t* test. **g** Transcript level of *SlPIF4* in the fruit of RNAi-*SlPIF4* transgenic lines as well as WT. **h** Transcript level of *SlCOMT2* in the fruit of RNAi-*SlPIF4* transgenic lines as well as WT. **i** Melatonin content in fruit of RNAi-*SlPIF4* transgenic lines as well as WT after red light treatment. Data in **g, h**, and **i** are represented as mean ± SEM ($n = 3$). In which 10–12 individual tomato fruits at the Br+ 3 stage were pooled as one biological replicate. The *P* values indicate the results from pairwise comparisons of one-way ANOVA tests. Different letters represent a significant difference at $P < 0.05$. Source data are provided as a Source Data file.

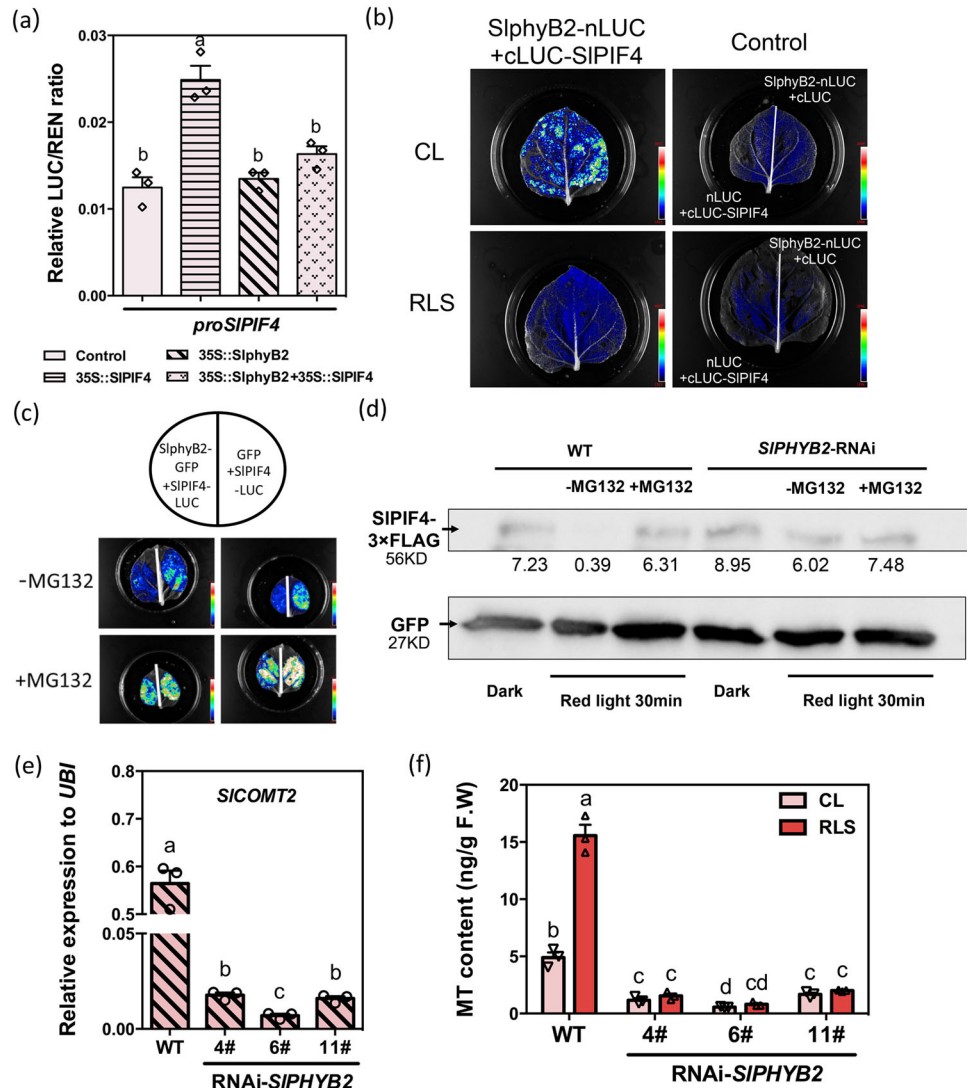

**Fig. 5 | Red light response of melatonin mediated by SlphyB2-SlPIF4-SlCOMT2.**
**a** The transcriptional regulation relationship between SlphyB2 and *SlPIF4*. The dual-LUC experiment proves that SlphyB2 can inhibit the self-activation of SlPIF4 on its own promotor. Data are represented as Mean ± SEM (*n* = 3). **b** Quantitative analysis of luminescence intensity showing the interaction between SlphyB2 and SlPIF4 in *Nicotiana benthamiana* leaves. SlphyB2 interacts with the SlPIF4 protein, but the interaction disappears under red light. **c** SlphyB2 can ubiquitously degrade SlPIF4, and MG132 prevents the degradation. **d** Western blot detection of ubiquitination degradation of SlPIF4 mediated by SlphyB2. The addition of MG132 will inhibit the degradation of SlPIF4 by red light in WT, while in the interference strains of *SlPHYB2*, the bands of SlPIF4 are not different. GFP acts as an actin ensure

consistent protein levels. This experiment was repeated independently two times with similar results. **e** Gene expression of *SlCOMT2* in the *SlPHYB2* interference lines. Samples were collected at Br+3. **f** Silence of *SlPHYB2* makes the plant no longer be induced by the red light to produce more melatonin. The content of melatonin in wild tomato fruit was induced and accumulated by red light, but decreased in RNAi-*SlPHYB2* lines, and was no longer induced by red light. Samples were collected at Br+3. Data in **e**, **f** are represented as Mean ± SEM (*n* = 3). In which 10–12 fruit collected from the same seedling were pooled as one biological replicate. For **a**, **e**, and **f**, the *P* values indicate the results from pairwise comparisons of one-way ANOVA tests. Different letters represent a significant difference at *P* < 0.05. Source data are provided as a Source Data file.

the G-Box element in the P2 site of *proCOMT2*, while P1 and P3 are invalid sites for SlPIF4 binding, which is consistent with the results found above (Fig. 4f).

Together, these data suggest that SlPIF4 can suppress the expression of *SlCOMT2* through interaction with the P2 site of *proSlCOMT2*. Under normal growth conditions, the expression of *SlCOMT2* was significantly upregulated in the RNAi-*SlPIF4* lines, together with significant induction of the MT content (Fig. 4g–i). After red light supplementation, although there was still significant induction of MT contents in the transgenic lines (27% and 16% increase, respectively), their MT content enhancement ratios were significantly lower than that of the WT fruit (63%) (Fig. 4i, S20). These data indicate that SlPIF4 is a negative regulator of MT biosynthesis and is involved in the red light-mediated regulation of MT biosynthesis.

## The SlphyB2-SlPIF4-SlCOMT2 module mediates red light-induced melatonin biosynthesis in tomato fruit

As one of the key plant phytochrome photoreceptors, phytochrome B2 (phyB2) plays an important role in red light response signaling[41–43]. In the MMN database[34], the expression of *SlPHYB2* in different developmental stages of tomato fruits is highly consistent with that of melatonin, while *SlPHYB2* and *SlPIF4* show opposite trends (Fig. S21). We first checked whether SlphyB2 can inhibit the expression of *SlPIF4*. The Dual-Luc assay using tomato protoplasts indicated that SlPIF4 could bind to its own promoter to achieve self-activation. Although SlphyB2 alone did not inhibit the activity of *proSlPIF4*, it inhibited the self-activation of *SlPIF4* (Fig. 5a).

Previous studies suggest that phyB2 activates the thermo-response by regulating PIF4 stability[44,45]. Firefly luciferase

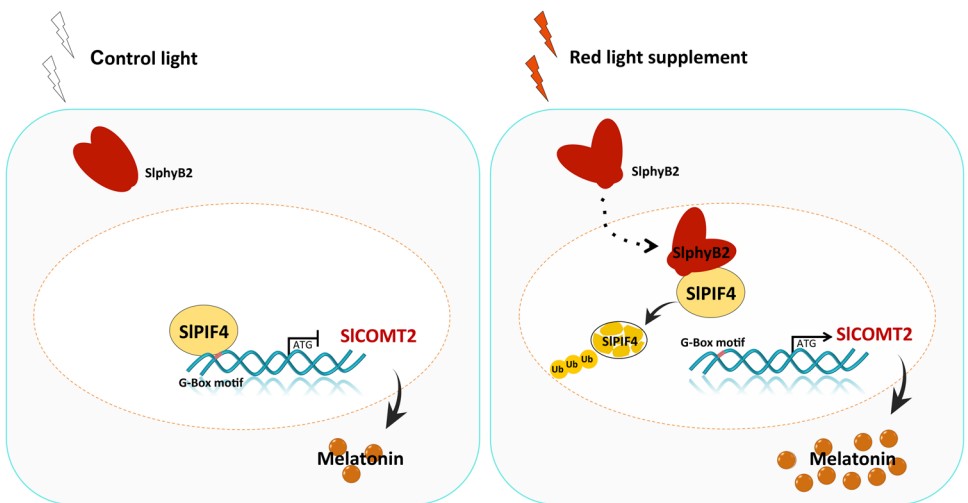

**Fig. 6 | Schematic representation of the molecular mechanism of red light-induced melatonin biosynthesis in tomato fruit.** SlphyB2 is activated under the red light supplement and can facilitate the degradation of SlPIF4 through the 26 S proteasome pathway, thus removing the inhibition of *SlCOMT2* by SlPIF4, leading to the accumulation of melatonin.

complementation imaging assays were performed to elucidate the interaction between SlphyB2 and SlPIF4 (Fig. 5b, S22). When *SlPIF4-LUC* was expressed together with *SlPHYB2* in tobacco leaves, the luciferase signal was significantly decreased. This inhibition can be removed by adding the proteasome inhibitor MG132 (Fig. 5c). This indicates that SlphyB2 might facilitate the degradation of SlPIF4.

To verify that SlphyB2 can regulate SlPIF4 at the protein level in vivo, we transiently overexpressed FLAG-tagged *SlPIF4* in both WT and RNAi-*SlPHYB2* tomato fruit. In the WT fruit, compared to fruit stored in the dark, the SlPIF4 protein content in agroinfiltrated fruit under light was significantly reduced. This phenotype can be effectively blocked by infiltrating the proteasome inhibitor MG132 into fruit (Figs. 5d, S23, S24). In the RNAi-*SlPHYB2* tomato fruit, however, the degradation of SlPIF4 under red light supplementation was effectively inhibited. These data indicate that SlphyB2 can regulate SlPIF4 stability via the 26 S proteasome pathway. Consequently, in the RNAi-*SlPHYB2* lines (Fig. S25), the expression level of *SlCOMT2* was inhibited, the melatonin content was significantly decreased, and the red light treatment was no longer effective (Fig. 5e, f, S26).

In summary, SlPIF4 negatively regulates melatonin biosynthesis in tomato fruit via direct inhibition of *SlCOMT2* expression. Under red light supplementation, the activation of SlphyB2 facilitates the degradation of SlPIF4 via the 26 S proteasome pathway. Therefore, the inhibition of *SlCOMT2* expression was released, and the biosynthesis of melatonin was enhanced (Fig. 6).

### Engineering melatonin-enriched tomatoes

To test whether this regulatory mechanism can be used for breeding tomato varieties with enhanced melatonin production, two gene-editing strategies were designed. One strategy is to directly knockout *SlPIF4* (Fig. 7a), and the other is to mutate the SlPIF4 recognition site on *proSlCOMT2* (Fig. 7b). Both methods significantly enhanced the production of melatonin (Fig. 7c). However, gene-editing targeting the SlPIF4 recognition site on *proSlCOMT2* can induce much stronger melatonin accumulation under normal growth conditions: Compared with WT, the melatonin content of the two CR-*slpif4* strains (12# and 15#) increased by approximately threefold. However, the melatonin content in the two CR-*proslcomt2* strains (5# and 8#) increased by 8.75- and 12.64-fold, respectively (Fig. 7c, S27).

## Discussion

Melatonin is an indoleamine compound found in all organisms from plants to animals[32,46]. Unlike animals, whose melatonin biosynthesis pathway has been thoroughly investigated[47,48], the melatonin biosynthesis pathway in most plants remains uncharacterized[9,49]. In this study, the biosynthetic pathway of melatonin in tomato was fully elucidated. We found that alternative melatonin biosynthesis routes coexist in tomato. One is through the "serotonin (**3**)−N-acetylserotonin (**5**)−melatonin (**6**)" route, in which SlASMT5 and SlCOMT2 are the key enzymes (Fig. 1c). The other route is the "serotonin (**3**)−5-methoxytryptamine (**4**)−melatonin (**6**)" route, in which SlASMT7 is the core enzyme (Fig. 1c).

As sessile photoautotrophic organisms, plants are constantly challenged by diverse external environmental conditions. To develop resistance capacity, plants produce various environment-induced metabolites, such as nutrients, antinutrients, and phytohormones[50,51]. In this study, we found that red light treatment at the fruit development stage can effectively induce the synthesis of melatonin in tomato fruit. Although there are alternative routes for melatonin biosynthesis in tomato (Fig. 1c), the red light-induced melatonin enhancement mainly relies on the "serotonin (**3**)−N-acetylserotonin (**5**)−melatonin (**6**)" route (Fig. 3b) via the activation of *SlCOMT2* and *SlASMT5* (Fig. 3a,c).

We further found that SlPIF4 can directly inhibit the expression of *SlCOMT2*. Under red light supplementation, the activation of SlphyB2 facilitates the degradation of SlPIF4. Therefore, the inhibition of *SlCOMT2* was released (Figs. 3–6). Previous studies have shown that PIFs are bHLH family transcription factors that can bind to photoreceptor phytochrome proteins (PHYs), and phytochrome can accelerate the degradation of the PIF-dependent 26 S proteasome by promoting the phosphorylation of PIFs under red light[41–43]. Studies in *Arabidopsis* show that PIF4 is a negative regulator of plant light signal transduction and can antagonize and regulate plant signal transduction[52,53]. The tomato phytochrome interaction factor PIF4 regulates tomato plant responses to temperature stress by integrating light and temperature hormone signals[54]. SlPIF4 has close homology with *Arabidopsis* AtPIF4, while AtPIF4 is not only a transcription factor necessary for the process of light signaling but can also positively regulate the synthesis of anthocyanins[55,56]. The joint cross-response of multiple environmental factors is the general trend of future research on plant growth and development and quality formation. Light and temperature often act on plants together, and PIF4, as an important

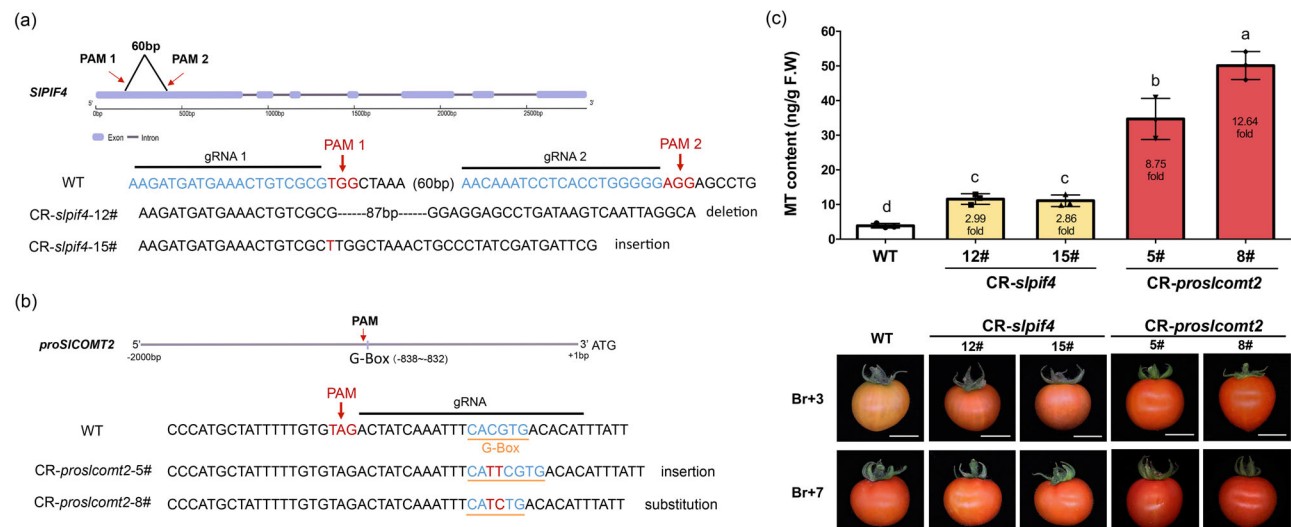

**Fig. 7 | Engineering of tomato germplasm with high melatonin content.**
**a** CRISPR/Cas9 target site design and sequencing results of gene editing for *SlPIF4*.
**b** CRISPR/Cas9 target site design and sequencing results of gene editing for the promoter of *SlCOMT2*. Cas9 recognize NG-PAM sequence. **c** Melatonin content of WT and T2 CR fruits at the Br+3 stage. Data is represented as Mean ± SEM ($n$ = 3).

10–12 tomato fruits from the same seedling were pooled as one biological replicate. The *P* values indicate the results from pairwise comparisons of one-way ANOVA tests. Different letters represent a significant difference at $P < 0.05$. Fruit phenotype of WT and CR fruits at the Br+3 and the Br+7 stage were also presented. Source data are provided as a Source Data file.

transcription factor of light and temperature signals, may be a link for further exploration of other regulatory genes and pathways. Indeed, we did find some other TFs, including members of the bHLH, bZIP, WRKY, and MYB families, in the Y1H screen library (Supplementary Data 1), which we will further investigate in subsequent studies. It has been reported that *HsfA1a* in tomato plants can promote the synthesis of melatonin to confer cadmium tolerance[57]. In cassava, MeHsf20, MeWRKY79 and MeRAV1/2 are able to induce melatonin production by binding to the promoters of melatonin biosynthesis genes[58]. However, most studies on the involved TFs are related to the stress response, and more TFs affecting melatonin synthesis need to be identified[32].

Recent studies have revealed that effective tomato metabolic engineering can be achieved by gene-editing targeting key biosynthesis-related genes[16,17]. Our data indicate that silencing and knocking out *SlPIF4* can significantly enhance the production of melatonin in tomato fruit (Fig. 4g–i), even under normal growth conditions. However, due to the vital functions of SlPIF4 in various signaling pathways[56,59], it is not wise to simply knock down/out this master regulator. Alternatively, we targeted the SlPIF4 recognition site in *proSlCOMT2* to design gene-editing strategies. By doing so, we can also significantly enhance melatonin production in tomato fruit. Previous studies have indicated that during tomato fruit ripening, DNA methylation is the key regulatory component[60,61]. The DNA methylation rate of SlCOMT2 was checked from the green stages to the ripening stages in our unpublished database. We found that at the green stage, *proSlCOMT2* was highly methylated (Fig. S28). Therefore, even without the inhibition of SlPIF4, the expression of *SlCOMT2* is low during green fruit stages in *slpif4* or *proslcomt2* mutants. In fact, this is the key advantage of gene editing for *proSlCOMT2*, which only removed SlPIF4 inhibition during the ripening stages without changing its expression pattern during other stages. Notably, compared to directly knocking out *SlPIF4*, the *proslcomt2* mutants had significantly higher melatonin production than the *slpif4* mutants (Fig. 7c), possibly due to other unknown TFs (Supplementary Data 1) interacting with the mutated G-box motif.

In summary, this study elucidated the full melatonin biosynthesis pathway in tomato fruit. We also uncovered the mechanism of red light induction of melatonin biosynthesis and successfully developed melatonin-enriched tomato varieties through gene editing. Our findings demonstrate that understanding the mechanisms by which environmental factors regulate key metabolic processes can be used to create nutrient-enriched crops.

## Methods

### Plant materials, growth conditions, and light treatments

Tomato (*Solanum lycopersicum* L. cv. MicroTom) seeds (purchased from Pan American Seed, Inc., Hillsborough, FL, USA) were grown in a standard greenhouse under a 16 h photoperiod (16 h light/8 h dark at 23 °C, relative humidity 70%). The light intensity indicated as PPFD (photosynthetic photon flux density), was set at 250 µmol m$^{-2}$ s$^{-1}$ above the plant canopy and maintained by adjusting the distance of 15 cm from the LEDs to the canopies. Red light refers to replacing 30% white light with red light, i.e., 30% red light at a wavelength of 657 nm and 70% white multiwavelength light, with white light as a control. The collected tissues were frozen in liquid nitrogen and stored at −80 °C until further investigation. Three biological replicates, each of which was a pooled sample of 10–12 individual fruits, were analyzed.

### Melatonin and metabolic intermediate extraction and analysis

The tomato tissues from three independent biological samples were ground into a fine powder and used for melatonin and metabolic intermediate measurements based on the AB Sciex QTRAP 6500 LC–MS/MS platform. In short, 200 mg tomato powder was extracted with 1.0 mL 80% aqueous methanol by ultrasonication for 20 min at 4 °C. The supernatants were transferred into new Agilent tubes after 10 min of centrifugation at 10,000 × *g* for LC–MS/MS analysis. The LC analytical conditions were as follows: samples were separated using a Hypersil Gold C18 column (100 × 2.1 mm, 1.9 µm; Thermo Fisher Scientific, USA) and the column temperature was set at 40 °C. The flow rate was 0.4 mL/min. The mobile phases were 0.1% formic acid (A) and acetonitrile (B), and the gradient was as follows: 5% B for 0.5 min; 5–95% B for 8 min; 95% B for 2 min followed by a decrease to 5% B for 0.1 min and re-equilibration of the column for 2.9 min with 5% B. The injection volume was 1.0 µL and the sampler temperature was set at 15 °C. Mass spectrometry was performed using an electrospray ionization (ESI) source. The source parameters in positive polarity were as follows: ion source gas 1: 25 psi; ion source gas 2: 60 psi; curtain gas: 40 psi; CAD gas: medium; temperature: 450 C; spray voltage: 5500 V.

Fragment XICs were extracted using SCIEX OS software (version 1.7). The same method was used for calibrating and quantifying the mass spectrum peaks of melatonin. Source data are provided as a Source Data file.

## Coexpression/coregulation identification and analysis

The Tomato Expression Atlas database[35] and the MMN database[34] were used for the preliminary identification of melatonin biosynthesis-related genes according to coexpression/coregulation analysis. Heat-maps created by R (v3.6.0) displayed for high-throughput analysis of the expression levels of the coexpressed genes.

## Plasmid construction and generation of transgenic lines

The subject sequence was introduced into the relevant vector by a homologous recombination system (ClonExpress® II One-Step Cloning Kit, C211, Vazyme) or restriction endonuclease reaction. pEAQ (for overexpression) and pTRV (pTRV1 and pTRV2 vectors, for virus-induced gene silencing, VIGS) were used for transient transformation. *pCAMBIA1306* (35 S::3×FLAG) was used for constitutive expression, and *pBWA(V)HS-RNAi* was used for RNA interference construction. *pHSbdcas9i* (for SlPIF4) *and pKSE401* (for proSlCOMT2) were used as the vector backbone for a one-step CRISPR/Cas9 binary constitutive system. A plasmid with the correct insertion was introduced into *Agrobacterium tumefaciens* strain EHA105. Source data are provided as a Source Data file.

## In vitro enzyme activity verification

The assay was performed according to the method described by Fu et al.[62,63]. The subject sequence was introduced into the *pDEST17* vector by the Gateway system. Methyltransferase and acetyltransferase were selected for enzyme activity verification, and heat shock transformation was carried out with *Escherichia coli* BL21 (DE3). Single colonies were selected from LB solid medium (50 μg/mL. ampicillin), and cultivated in 200 mL LB liquid medium with corresponding resistance at low speed for 3–5 h at 37 °C. The positive strains were obtained by polymerase chain reaction. Subsequently, 20 μL of the bacterial solution was added to LB medium containing antibiotics and incubated overnight at 37 °C until the $OD_{600}$ reached 0.5-1.0. Isopropyl β-D-1-thiogalactopyranoside (IPTG) was added to a final concentration of 0.5-1.0 mM and induced at 28 °C for 8 h. SDS polyacrylamide gel electrophoresis (SDS–PAGE) was performed to determine whether the protein was expressed.

The *Escherichia coli* liquid with the target protein was centrifuged at 4 °C at $5000 \times g$ for 10 min. Cells were re-suspended in lysis buffer (25 mM Tris-HCl 8.0, 150 mM NaCl, 0.5 mM tris-phosphine) with 1 mM ATP and 1 mM PMSF. The cells were broken by a disruptor and centrifuged at 4 °C at $20,000 \times g$ for 30 min. The supernatant was collected and proteins were purified with $Ni^{2+}$-NTA column (Qiagen, Germany) according to the manufacturer's guidance. The concentration of purified protein was determined by the BCA Protein Assay Kit (Sangon Biotech, China). The enzyme assay was performed using a 100 μL reaction system containing 1 μg of purified enzyme, 20 μM of each substrate in 1× PBS (137 mM NaCl, 2.7 mM KCl, 10 mM $Na_2HPO_4$ and 2 mM $KH_2PO_4$, pH = 7). The mixture was incubated at 30 °C for 60 min and the reaction was stopped by adding 400 μL of methanol. Then, the samples were centrifuged at 4 °C at $20,000 \times g$ for 10 min, and the supernatant was used for mass spectrometry.

## Subcellular localization

The full-length coding region without the termination codon was amplified with 35 S::GFP (*pCAMBIA1302*). It was then transformed into *Arabidopsis* protoplasts after incubation for 12 h at 28 °C. *Arabidopsis* leaves (3 weeks) were used for protoplast separation. A vacuum was applied for 20 min after enzymatic hydrolysis, followed by the addition of W5 and resuspension in an ice bath. Then, 200 μL of protoplast was added to the target plasmid. The protoplasts were placed in the dark at 24 °C for 20 h after 40% PEG-mediated transformation. A confocal laser scanning microscope (LSM800, Zeiss) was used for GFP fluorescence detection.

## Construction and screening of the yeast library

Five hundred micrograms of high-quality total RNA were extracted from tomato tissues, and Gateway technology was used for yeast library construction. The cDNA library was prepared by Yuanbao Bio-tech (Nanjing, China). SD/-His-Leu-Trp dropout 3AT culture screening plates and the Y187 yeast strain were used to screen the yeast library. Each obtained more than 600 clones, and high-throughput sequencing was performed after colony collection. The raw data was transferred to fasta using fq2fa and then aligned to the tomato iTAG4.0, the paraments were outfmt 6, e value was $1e^{-3}$.

## Yeast one-hybrid assays

The promoter fragments were amplified, cloned, and inserted into the *pLacZi* vector, and the CDS of *SlPIF4* was fused to *pB42AD*. The constructs were then transformed into the yeast strain EGY48, and yeast cells were inoculated on a selective medium for 3 days at 28 °C and transferred to SD/-Ura-Trp medium. Yeast colonies turned blue with X-gal if there was an interaction between the factors.

## Transient dual-luciferase reporter assay

The fragment of the *SlCOMT2, SlASMT5 or SlPIF4* promoter was cloned and inserted into the *pGreenII 0800-LUC* vector. *Agrobacterium tumefaciens* strain GV3101 harboring targeted fragments was grown in infiltration medium (2 mM $Na_3PO_4$, 50 mM MES, and 100 mM acetosyringone) to an $OD_{600}$ of 0.5 and then introduced via a syringe into the leaves of a 4–5-week-old *Nicotiana benthamiana* plant. After 48–96 h, a CCD camera was used to observe luciferase activity. Tomato leaves (2–3 weeks) were used for protoplast separation. After enzymatic hydrolysis, a vacuum was applied for 30 min, followed by the addition of W5 and resuspension in an ice bath. Then, 200 μL of protoplast was added to the target plasmid. After 40% PEG-mediated transformation, the protoplasts were placed in a dark environment at 24 °C for 20 h. The Dual-Luciferase Reporter Assay System E1960 (Promega, cat. #e1910, Madison, USA) was used to measure the fluorescence intensity of luciferase and renilla (REN). The relative LUC/REN ratios were used to represent the activity of the promoters.

## ChIP–qPCR assay

The transgenic Line 35 S::FLAG-*SlPIF4* was assessed by ChIP–qPCR assays. Six grams of the breaker stage fruit tissues were sliced into small pieces and immersed in the crosslinking buffer (0.4 M Sucrose, 10 mM Tris-HCl pH 8.0, 0.1% β-mercaptoethanol, 100 mM PMSF, 1% formaldehyde and 1× protease inhibitor cocktail (Roche) and vacuum infiltrated three times, for 10 min each time. Glycine was added to a final concentration of 0.125 M and samples were vacuum infiltrated for an additional 5 min. Samples were washed and frozen by liquid Nitrogen and ground into fine powder. Chromatin isolation was performed using the Honda buffer (0.44 M sucrose, 1.25% (wt/vol) Ficoll, 2.5% (wt/vol) Dextran T40, 20 mM HEPES (pH 7.4), 10 mM $MgCl_2$, 0.5% (vol/vol) Triton X-100, 1 mM DTT, 1× protease inhibitor cocktail). Samples were re-suspended with 25 ml Honda buffer on ice for 5 min and filtered through two layers of Miracloth. Pellets were collected after centrifugation at $2000 \times g$ at 4 for 10 min and re-suspended again with 1 mL Honda buffer, repeating the centrifugation and resuspension at least three times. Nuclei were isolated in Nuclei Lysis Buffer (50 mM Tris-HCl pH 8, 10 mM EDTA, 1% SDS, 1 mM PMSF, and 1× Protease Inhibitors) and DNA was sheared into ~250–750 bp via sonication. ChIP was performed using monoclonal anti-FLAG protein antibody (Sigma-Aldrich, F1804). The

immunoprecipitation and DNA isolation/purification were done using the EpiTect ChIP OneDay Kit (Qiagen, Germany) according to the manufacturer's instructions. The primers used for the qPCR assay are listed in Table S3, and each was repeated at least three times.

### Electrophoretic mobility shift assay (EMSA)
The fusion proteins of SlPIF4 were generated through prokaryotic expression in vitro. The CDSs of *SlPIF4* were cloned and inserted into the *PGEX-5T* vector containing a GST target and expressed in *Escherichia coli* strain BL21. IPTG was used to induce protein production. The MagneGST™ Pull-Down System (Promega, USA) was used for protein purification, and the LightShift™ Chemiluminescent EMSA Kit (Thermo Fisher, USA) was used for the subsequent EMSAs. Unlabeled probes were used for probe competition. Then, it was loaded onto a prerun native 6.5% polyacrylamide gel with TBE buffer as the electrolyte. After electroblotting onto a nylon membrane (Millipore, Germany) and UV crosslinking (2000 J for 5 min), the membrane was incubated in blocking buffer for 30 min and rinsed in washing buffer. A CCD camera was used to visualize the chemiluminescent signal.

### Floated-leaf luciferase complementation imaging assay
To investigate whether SlphyB2 interacts with SlPIF4 in vivo, we used the *pCAMBIA1300-cLUC* and *pCAMBIA1300-nLUC* vectors for the FLuCI assay. SlphyB2 was fused to the C-terminal fragment of luciferase (cLUC), while SlPIF4 was fused to the N-terminal fragment of luciferase (nLUC). The interactions between nLUC and SlphyB2-cLUC as well as SlPIF4-nLUC and cLUC were used as negative controls. The final constructs were transformed into *Agrobacterium tumefaciens* strain GV3101, and different combinations of plasmids were co-infiltrated into *Nicotiana benthamiana* leaves. After incubation in the dark for 12–14 h and then in light for 48 h, the tobacco leaves were sprayed with 100 mM D-luciferin and kept in the dark for 5–10 min, then photographed with a CCD camera.

### Validation of ubiquitination degradation
The ubiquitination and degradation of SlPIF4 by SlphyB2 were validated with the help of the proteasome inhibitor MG132 (Beyotime, S1748), which can effectively block the proteolytic activity of the 26 S proteasome complex. 1 mL 80 μM MG132 (10 mM MgCl$_2$, 80 mM MG132) and its reference solution were injected 6 h before collection. SlPHYB2 was cloned and inserted into 35 S::GFP (pCAMBIA1302), while SlPIF4 was constructed with a luciferase vector. A CCD camera was used to observe luciferase activity in tobacco leaves. A plasmid containing both FLAG and GFP labels for SlPIF4 was used for detecting co-infiltrating control protein to make sure the degradation. Anti-GFP protein antibody bought from Abcam (ab290). For western blotting, 30 DPA tomato fruits were selected from RNAi-*SlPHYB2* and wild-type plants, and infection solution was injected from the bottom of the fruit until liquid leached at the stem. The infected fruits were incubated in the dark for 24 h, followed by 3 days of dark cultivation. Half of the plants injected with MG132 or its reference solution were treated with red light for 30 min. BCA was used to determine the total protein concentration. SDS–PAGE electrophoresis was performed with consistent protein content in each sample. Source data are provided as a Source Data file.

### Total RNA isolation and qRT–PCR analyses
Samples were harvested and ground into a fine powder using liquid nitrogen. Total RNA was extracted using RNAiso reagent (BIOFIT, RN33050) as recommended by the manufacturer. One microgram of RNA was used for first-strand cDNA by the PrimeScript™ RT reagent Kit containing gDNA eraser (Takara, Kusatsu, Japan). qRT–PCR was performed using the Bio-Rad CFX384 Real-Time System according to the manufacturer's instructions. The relative expression level of each gene was calculated using the ΔCt method as described previously[34],

and *SlUBI* was used as an internal control. Average values were calculated by three biological replicates (*n* = 3). One biological replicate is the pool of 10–12 samples.

### Statistical analysis
At least three biological replicates were included in the data, and the statistical significance of differences was determined by ANOVA followed by post hoc Tukey's test or Student's *t* test (GraphPad Prism version 8).

### Accession numbers
The accession numbers of genes are as follows: *SlTDC1* (Solyc07g054860, https://solgenomics.net/locus/30030/view), *SlTDC2* (Solyc07g054280, https://solgenomics.net/locus/29973/view), *SlT5H* (Solyc09g014900, https://solgenomics.net/locus/33811/view), *SlSNAT* (Solyc10g074910), *SlASMT7* (Solyc06g064500, https://solgenomics.net/locus/37357/view), *SlASMT5* (Solyc03g097700, https://solgenomics.net/locus/18964/view), *SlCOMT2* (Solyc10g085830, https://solgenomics.net/locus/38127/view), *SlPHYB2* (Solyc05g053410, https://solgenomics.net/locus/25083/view), *SlPIF4* (Solyc07g043580, https://solgenomics.net/locus/29516/view).

### Reporting summary
Further information on research design is available in the Nature Portfolio Reporting Summary linked to this article.

## Data availability
Data supporting the findings of this work are available within the paper and its Supplementary Information files. A reporting summary for this Article is available as a Supplementary information file. The datasets and plant materials generated and analyzed during the current study are available from the corresponding author upon request. The source data underlying Figs. 1d, 2d–g, 3c, d, 4a, b, and 6, Figs. 2a, 3b, c, 4i, and 5f, 7c, as well as Supplementary Figs. S12, S13, S15, S16, S19a, S20, S23, S24a, S25a, S26, S27 are provided as a Source Data file. Source data are provided with this paper.

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

## Acknowledgements

This work was financially supported by grants from the National Key Research and Development Program of China (2022YFF1001900, Y.Z.), the National Natural Science Foundation of China (32200260, Z.X.Z.), the Natural Science Foundation of Sichuan Province, China (2023NSFSC1991, Y.Z.), the China Postdoctoral Science Foundation Funded Project (2020M673207, Z.X.Z.) and the Sichuan University Postdoctoral Science Foundation Funded Project, China (2020SCU12061, Z.X.Z.). We acknowledge the Special Fund for Fundamental Research Funds for the Central Universities (2023SCUD0003, Y.Z.). We acknowledge Dr. Hsihua Wang from the Center of Metabolomics and Proteomics in the College of Life Science, Sichuan University, for technical support in metabolic analysis.

## Author contributions

Y.Z. and Z.X.Z. conceived and designed the experiments; Z.X.Z., X.Z., Y.T.C., and W.Q.J. performed most of the experiments; J.Z., J.Y.W. and Y.J.W. provided technical support; X.Y. provided technical support on light treatment; S.C.W. and M.C.L. provided conceptual advice; W.Q.J. and X.Z. contributed to plant transformation; Z.X.Z., X.Z. and Y.Z. analyzed the data and wrote the manuscript with inputs from all authors.

## Competing interests

The authors declare no competing interests.
