## [Peer Review File · Nature Communications]

Understanding the mechanism of red light-induced melatonin biosynthesis facilitates the engineering of melatonin-enriched tomatoesREVIEWER COMMENTS

Reviewer #1 (Remarks to the Author):

The manuscript "Understanding the mechanism of red light-induced melatonin biosynthesis facilitates the engineering of melatonin-enriched tomatoes" presents a new characterization of the latter genes of melatonin biosynthesis in tomato, and present the production of transformed tomato with increase melatonin or nutraceutical and agronomic purposes. However, the information and documentation presented in this manuscript are not detailed and clear enough, and sometime completely missing, to support the authors conclusions. For instance, in the "Plant materials, growth conditions and light treatments" the sentence "Three independent biological repeats were performed for each experiment" doesn't not really explain what are exactly the three independent biological repeats in the context of the many experiments of this paper. What are exactly the experimental units? Three pieces of fruits? Three whole fruits (for tomato) or leaves (for Nicotiana)? Three pools of fruits or leaves? Collected from how many plants? The generic and recurrent sentence "Data are represented as mean \pm SD (n=3)" it is not a clear indication of sample type, sample size and sample representativeness for each experiment.

The manuscript is based on the functional characterization of the genes and enzymes of melatonin biosynthetic pathway, the nature of which was verified by the authors through the detection of the product metabolites after overexpression or silencing of the biosynthetic genes, and after the in vitro enzymatic reaction using the recombinant proteins produced in E.coli.

But:

-the LC-MS procedure for metabolite detection is completely missing: the paragraph "Melatonin and metabolic intermediates extraction and analysis" say only that measurement was "based on the AB SciexQTRAP 6500 LC-MS/MS platform": not sufficient for a paper that is entirely based on these key determinations. Strongly relevant information could not be skipped.

-The procedure of recombinants proteins production in E.coli are completely skipped, as well as the in vitro enzymatic assays procedure.

Also other procedures (as the protoplasts transfection) are completely missing.

Also the figures and their captions are often not enough explanatory: for instance, figure 2b, which should show a key evidence of the enzymes roles, is entirely not explained.

The manuscript is sometime confused, with mismatches between documentation and text: for instance, the "Arabidopsis protoplasts" of figure S5 are "tomato protoplasts" in the text; in figure S9 only the RNAi of SIASMT5 is shown to reduce methoxytryptamine, while in the text the authors say that "only the silencing of SIASMT7 can significantly reduce the contents of 5-methoxytryptamine".

LC-MS cannot be used for absolute metabolite quantification simply using the calibration curve with external standards, due to the strong matrix effects of the ionization sources. The use of other methods (for instance internal deuterated standards) is necessary.

Reviewer #2 (Remarks to the Author):

Authors defined tomato fruit-specific melatonin biosynthetic genes using transcriptome data, transient expression data, and gene silenced data. After defining fruit-specific melatonin biosynthetic genes, authors showed that red light increases melatonin biosynthesis by increasing the expression of SICOMT2, which can be explained by phyB2-SIPIF4-G-box in

proSICOMT2 regulatory path. Authors then engineered melatonin-rich tomato by removing a G-box in proSICOMT2. I have a few comments.

1. EMSA data presented in Fig 4e is not clear. Even 250x cold probe does not seem to compete very well. Besides, there is a black smearing at the position of SIPIF4 even in the first lane. I think EMSA data in Fig 4e should be updated.
2. Fig 4d and 4f do not match with figure legends.
3. The N-terminal region of phyB forms a pseudoknot structure, thus, the fusion of other protein at its N-terminus is not tolerable. Authors fused cLUC domain to the N-terminus of SIPHYB2 to perform split luciferase assay (Fig 5b). Though the interaction between phyB and PIF4 is expected, I think authors should not fuse cLUC to the N-terminus of SIPHYB2. How about fusing it to the C-terminus? The fusion of a protein at the C-terminus of phyB has shown to be highly tolerable.
4. Data in Fig 5d need to be strengthened by detecting co-infiltrating control protein. The data were made by agro-infiltrated FLAG-SIPIF4 and injected MG132. To make sure that the difference in SIPIF4 protein level is due to the degradation rather the complication associated with agro-infiltration, authors may want to co-infiltrate non-degradable protein, say GFP, and normalize SIPIF4 protein level by GFP level.
5. Red light-treated tomato (Fig 3a) and *slpif4* mutant tomato (Fig 7c) produce similar amounts of melatonin, which are much lower than a G-box-mutated tomato (Fig 7c). G-box is known to be a binding site of various transcription factors including various bHLH TFs and bZIP TFs, strongly suggesting the presence of other important regulators. I recommend to modify too simplistic model in Fig 6 and discuss properly in the discussion section. BTW, what is the sky blue line where Pr is located in Fig 6?
6. Tomato green fruit is suggested to be self-shading due to thick green flesh, which protects the degradation of PIFs. If SI-phyB1/SI-PIF4/G-box at proSI-COMT2 is the major axis of regulation, it is expected that SI-COMT2 is expressed constitutively or from early stage in *pif4* and *g-box* mutants. Is it the case? This will strengthen authors' claim of the central importance of SI-phyB1/SI-PIF4/G-box at proSI-COMT2.
7. All methods should be written in detail. For example, authors wrote that they injected MG132 (10 mM MgCl₂, 50 μM MG132) 6 h before the collection. But how much? Please go through the method section and describe various protocols more carefully.

Below are more less editorial points.

8. Please include at least a representative original metabolic profile for each in a supplementary figure. Currently, authors provided only LC/MS running profiles for standard chemicals, thus, it is difficult to judge authors' claims.
9. Authors wrote (n=3) for metabolite analysis and RT-qPCR analysis. I presume they are from three different tomato plants. Please indicate clearly what n=3 means in figure legends or method sections.
10. Authors are using 'melatonin synthase' genes indicate melatonin biosynthetic genes. Since synthase refers a specific type of enzyme, I suggest not to use synthase in this context.
11. Conventionally, non-italic capital lettered PHYB refers apoprotein, while phyB refers a holoprotein. Since authors are likely dealing with a holoprotein, please use phyB instead of PHYB.
12. Please indicate the line numbers in Fig S7 to relate them with S8. Same for Fig S12, S14 etc.
13. Fig S11 is incomprehensible. Please label each lane properly and write down detailed figure legends.
14. All suppl figures supporting the main figure should be cited in main figure legends and

vice versa. For example, Fig S15 seems to be associated with Fig 5d. Please cite each other in figure legends. An arrow in Fig S15 is misleading: the arrow indicates as if a band in SDS gel is PIF4. Authors may rather include the full immunoblot image of Fig 5d in supple Fig S15. I'm not quite sure the utility of SDS PAGE of plant cell extracts. Please also see #4.

We sincerely thank the editor and all reviewers for their valuable feedback that we have used to improve the quality of our manuscript. According to your suggestions, we have made extensive corrections to our MS; the detailed corrections are listed below. Our response is given in blue text and changes/additions to the manuscript are shown in the marked text.

Reviewer #1 (Remarks to the Author):

The manuscript "Understanding the mechanism of red light-induced melatonin biosynthesis facilitates the engineering of melatonin-enriched tomatoes" presents a new characterization of the latter genes of melatonin biosynthesis in tomato, and present the production of transformed tomato with increase melatonin for nutraceutical and agronomic purposes. However, the information and documentation presented in this manuscript are not detailed and clear enough, and sometime completely missing, to support the authors conclusions. For instance, in the "Plant materials, growth conditions and light treatments" the sentence "Three independent biological repeats were performed for each experiment" doesn't not really explain what are exactly the three independent biological repeats in the context of the many experiments of this paper. What are exactly the experimental units? Three pieces of fruits? Three whole fruits (for tomato) or leaves (for Nicotiana)? Three

pools of fruits or leaves? Collected from how many plants? The generic and recurrent sentence "Data are represented as mean +/-SD (n=3)" it is not a clear indication of sample type, sample size and sample representativeness for each experiment.

We have added detailed descriptions of sample collection and statistics in the corresponding place of the methods and legends section. Unless specifically mentioned, three biological replicates were calculated for most experiments. For T0 transgenic plants, each individual fruit from the same seedlings is recognized as one biological replicate. Please see the figure legends of Fig S11, S19 and S26. For WT and T1/T2 plants, one biological replicate is the pool of 10-12 fruit from the same seedling. Please see the figure legends of Fig 1c, 2a, 3a-c, 4d, 4g-i, 5e-f, 7c, S7 and S14.

The manuscript is based on the functional characterization of the genes and enzymes of melatonin biosynthetic pathway, the nature of which was verified by the authors through the detection of the product metabolites after overexpression or silencing of the biosynthetic genes, and after the in vitro enzymatic reaction using the recombinant proteins produced in E.coli. But:-the LC-MS procedure for metabolite detection is completely missing: the paragraph "Melatonin and metabolic intermediates extraction and analysis" say only that measurement was

"based on the AB SciexQTRAP 6500 LC-MS/MS platform": not sufficient for a paper that is entirely based on these key determinations. Strongly relevant information could not be skipped.

Detailed methods for the determination of melatonin and other intermediate products had also been added in the Materials and Methods section (Lines 435-444): "ACCUCORE C30 chromatographic column was used with the mobile phase of acetonitrile (solvent A)-methanol (solvent B)-ultrapure (solvent C) water (v/v/v). The column temperature was set to 18°C and the injection volume was 2µL. The gradient elution procedure with 1mL/min flow velocity was as follows, time (1, 2, 4.5, 7.5, 8, 10 min)/mobile phase (90%A-10%C, 100%A, 85%A-15%B, 100%A, 90%A-10%C, 90%A-10%C). Fragment XICs were extracted using SCIEX OS software (version 1.7). And the same method was used for calibrating and quantifying the mass spectrum peaks of melatonin. "

The procedure of recombinants proteins production in E.coli are completely skipped, as well as the in vitro enzymatic assays procedure. Also other procedures (as the protoplasts transfection) are completely missing.

We have added a detailed description of enzyme activity detection in the materials and methods section (Lines 464-486, Fu, et al., 2021, 2022):

"The assay was performed according to the method described by Fu et

al. The subject sequence was introduced into the pDEST17 vector by the Gateway system. Methyltransferase and acetyltransferase were selected for enzyme activity verification, and heat shock transformation was carried out with *Escherichia coli* BL21. The single colonies were selected and cultivated in LB liquid medium with corresponding resistance at low speed for 3-5 hours at 37 °C. The positive strains were obtained by polymerase chain reaction. Subsequently, 20 µL bacterial solution was taken to the LB medium containing antibiotics and incubated overnight at 37 °C until the OD600 reached 0.5~1.0. IPTG was added to a final concentration of 0.5~1.0 mM and induced at 28 °C for 8 hours. SDS polyacrylamide gel electrophoresis (SDS-PAGE) was performed to determine whether the protein was expressed.

The *Escherichia coli* liquid with the target protein was centrifuged at 4 °C at 5,000×g for 10 min. The collected solution was resuspended with 10 mL 1X PBS buffer, which mainly composed of Na₂HPO₄ and KH₂PO₄. An enzyme activity reaction was taken after treated by ultrasonic wave. Adding 400µL methanol to stop the reaction after incubation at 30°C for 1 hour. Then centrifuged at 4°C at 20,000×g for 10 minutes, and the supernatant was used for mass spectrometry.”

The protoplast procedures have been also added in lines 515-524: “2-3 weeks of tomato leaves were used for protoplast separation. After

enzymatic hydrolysis, vacuum was applied for 30 minutes, followed by the addition of W5 and resuspended in an ice bath. Add 200 μ L protoplast to the target plasmid. After 40% PEG-mediated transformation, the protoplasts were placed in a dark environment at 24°C for 20 h. The Dual-Luciferase Reporter Assay System (Promega, cat. #e1910, Madison, USA) was used to measure the fluorescence intensity of luciferase and renilla (REN). The relative LUC/REN ratios were used to represent the activity of the promoters. ”

Reference:

Fu, R., Zhang, P., Jin, G., et al. Versatility in acyltransferase activity completes chicoric acid biosynthesis in purple coneflower. *Nature Communications*, 2021, 12: 1563.

Fu, R., Zhang, P., Jin, G., et al. Substrate promiscuity of acyltransferases contributes to the diversity of hydroxycinnamic acid derivatives in purple coneflower. *The Plant Journal*, 2022, 118: 802-813.

Also the figures and their captions are often not enough explanatory: for instance, figure 2b, which should show a key evidence of the enzymes roles, is entirely not explained.

We have further explained Fig2b in its caption: “*In vitro* enzyme activity verification of key structural genes. Different proteins were incubated with different substrates (serotonin, N-acetylserotonin and 5-methoxytryptamine, respectively) to detect the production of

N-acetylserotonin, 5-methoxytryptamine, and melatonin, respectively.

‘EV’ indicates the empty vector for negative control.”

We have also added the protein purification process in the attached figure (Fig. S13) and revised methods for enzyme activity detection (Lines 464-486).

Fig. S13 The purification of SICOMT2, SIASMT7, SIASMT5 and SISNAT. (associated with Fig. 2b).

The manuscript is sometime confused, with mismatches between documentation and text: for instance, the "Arabidopsis protoplasts" of figure S5 are "tomato protoplasts" in the text;

We have corrected this typo. We used *Arabidopsis* protoplasts for protein localization assay, please see in lines 515-524 and new Fig. S9 (previously Fig. S5).

in figure S9 only the RNAi of SIASMT5 is shown to reduce methoxytryptamine, while in the text the authors say that "only the silencing of SIASMT7 can significantly reduce the contents of

5-methoxytryptamine".

We apologize for this typo. When labeling Fig. S9, we used the initial gene names in our group. The ASMT5.1 in S9 is the current article's ASMT7, and ASMT7.1 is the current article's ASMT5. Please see new Fig S14.

LC-MS cannot be used for absolute metabolite quantification simply using the calibration curve with external standards, due to the strong matrix effects of the ionization sources. The use of other methods (for instance internal deuterated standards) is necessary.

In previous studies, the detection of melatonin was mostly carried out by external standard methods. It provides reliable results and is widely accepted by the society of plant melatonin research (Pandi-Perumal, et al., 2008; Tan, et al., 2012; Gomez, et al., 2013; Arnao, et al., 2018; Li, et al., 2020). In addition, we have added the mass spectra (MSII spectrum) in supplemental Fig. S4 and extracted ion chromatogram (XIC) in supplemental Fig. S6, S12, S15, S16, S20, S26, S27.

References

- Pandi-Perumal, S., Trakht, L., Srinivasan, V., et al., Physiological effects of melatonin: role of melatonin receptors and signal transduction pathways. *Progress in neurobiology*, 2008, 85 (3): 335-353.
- Tan, D., Hardeland, R., Manchester, L., et al., The changing biological

roles of melatonin during evolution: from an antioxidant to signals of darkness, sexual selection and fitness. *Biological reviews*, 2012, 87(2): 334-355.

Gomez, F., Hernández, I., Martínez L., et al. Analytical tools for elucidating the biological role of melatonin in plants by LC-MS/MS. *Electrophoresis*, 2013, 34(12): 1749-1756.

Arnao, M. and Hernández-Ruiz, J. Melatonin: a new plant hormone and/or a plant master regulator? *Trends in plant science*, 2018, 23(10): 811-823.

Li, D., Guo, Y., Zhang, D., et al. Melatonin represses oil and anthocyanin accumulation in seeds. *Plant Physiology*, 2020, 183(3): 898-914.

Fig. S4 The mass spectra (MSII spectrum) of melatonin and its intermediates detected by LC/MS.

Fig. S6 The extracted ion chromatogram (XIC) of melatonin and its intermediates detected by LC/MS in instantaneously transformed samples. (associated with Fig. 1c).

Fig. S12 The extracted ion chromatogram (XIC) of melatonin detected by LC/MS in stable transgenic tomato. (associated with Fig. 2a).

Fig. S15 The extracted ion chromatogram (XIC) of melatonin detected by LC/MS in wild-type and transgenic tomato fruit under control light (CL) and red light supplement (RLS). (associated with Fig. 3c).

Fig S16 The extracted ion chromatogram (XIC) of N-acetylserotonin and 5-methoxytryptamine detected by LC/MS in wild-type tomato fruit under control light (CL) and red light supplement (RLS). (associated with Fig. 3b).

Fig. S20 The extracted ion chromatogram (XIC) of melatonin detected by LC/MS in wild-type and RNAi-*SIP4* transgenic tomato fruit under control light (CL) and red light supplement (RLS). (associated with Fig. 4i).

Fig. S26 The extracted ion chromatogram (XIC) of melatonin detected by LC/MS in wild-type and RNAi-*SlphyB2* transgenic tomato fruit under control light (CL) and red light supplement (RLS). (associated with Fig. 5f).

Fig. S27 The extracted ion chromatogram (XIC) of melatonin detected by LC/MS in wild-type and T2 CRISPR (CR) tomato fruit. (associated with Fig. 7).

Reviewer #2 (Remarks to the Author):

Authors defined tomato fruit-specific melatonin biosynthetic genes using transcriptome data, transient expression data, and gene silenced data. After defining fruit-specific melatonin biosynthetic genes, authors showed that red light increases melatonin biosynthesis by increasing the expression of SICOMT2, which can be explained by phyB2-SIPIF4-G-box in proSICOMT2 regulatory path. Authors then engineered melatonin-rich tomato by removing a G-box in proSICOMT2. I have a few comments.

1. EMSA data presented in Fig 4e is not clear. Even 250x cold probe does not seem to compete very well. Besides, there is a black smearing at the position of SIPIF4 even in the first lane. think EMSA data in Fig 4e should be updated.

We have reconducted this experiment. The EMSA data in Fig 4e have been updated.

Fig. 4e EMSA of SIPIF4 binding to the P2/mP2 fragment. SIPIF4

binds to the P2 fragment of *proSICOMT2*, while the mutant of P2 (mP2) does not present binding. '+' indicates presence; and '-' indicates absence.

2. Fig 4d and 4f do not match with figure legends.

Thank you! The description of figure legends has been corrected.

3. The N-terminal region of phyB forms a pseudoknot structure, thus, the fusion of other protein at its N-terminus is not tolerable. Authors fused cLUC domain to the N-terminus of SIPHYB2 to perform split luciferase assay (Fig 5b). Though the interaction between phyB and PIF4 is expected, I think authors should not fuse cLUC to the N-terminus of SIPHYB2. How about fusing it to the C-terminus? The fusion of a protein at the C-terminus of phyB has shown to be highly tolerable.

Thank you! We have reconducted this experiment by fusing nLUC to the C-terminus of SlphyB2 and the detection was taken by Floated-leaf Luciferase complementation imaging assay. We replaced Fig. 5b with the following right figure and included the original figure (left) as a supplementary data (Fig. S22).

Left, new Fig. 5b Quantitative analysis of luminescence intensity showing the interaction between SphyB2 and SIPIF4 in *Nicotiana benthamiana* leaves. Right, original Fig. 5b now supplemental Fig. S22.

4. Data in Fig 5d need to be strengthened by detecting co-infiltrating control protein. The data were made by agro-infiltrated FLAG-SIPIF4 and injected MG132. To make sure that the difference in SIPIF4 protein level is due to the degradation rather the complication assicauted with agro-infiltration, authors may want to co-infiltrate non-degradable protein, say GF, and normalize SIPIF4 protein level by GFP level.

Response: Thank you! We constructed a plasmid containing both FLAG and GFP labels for *SIPIF4*, and indeed the protein was indeed degraded by detecting the GFP label. We have added the data as Fig. 5d and Fig. S23.

Fig. Si Red light response of the degradation of SIPIF4 by *SlphyB2*. (a) Constructed plasmid map. (b) Western blot detection of ubiquitination degradation of SIPIF4 mediated by *SlphyB2*.

Right, new Fig. 5d; Left, supplemental Fig. S23.

5. Red light-treated tomato (Fig 3a) and *slpif4* mutant tomato (Fig 7c) produce similar amounts of melatonin, which are much lower than a G-box-mutated tomato (Fig 7c). G-box is known to be a binding site of various transcription factors including various bHLH TFs and bZIP TFs, strongly suggesting the presence of other important regulators. I recommend to modify too simplistic model in Fig 6 and discuss properly in the discussion section. BTW, what is the sky blue line where Pr is located in Fig 6?

Response: Indeed, we did find some other bHLH TFs in the Y1H, it is possible that additional TFs might bind to this G-box motif, which we will further investigate in the following studies. We also updated the schematic model in Fig. 6. As the roles of other TFs are under investigation, we can only include the current PIF4-COMT2 model in the figure. However, we do emphasize that there are other possible TFs in the discussion, please see lines 372-375: "And indeed, we did find some other TFs including bHLH, bZIP, WRKY, MYB, etc families in the Y1H screen library (Supplementary file 1), which we will further investigate in the following studies". As well as line 401-404: "Notably, compared to directly knocking out PIF4, the proslcomt2 mutations have significantly higher melatonin production than the pif4 mutants (Figure 7c), this was possibly due to other unknown TFs (Supplementary file 1) interacting with the mutated G-box motif."

6. Tomato green fruit is suggested to be self-shading due to thick green flesh, which protects the degradation of PIFs. If SI-phyB1/SI-PIF4/G-box at proSI-COMT2 is the major axis of regulation, it is expected that SI-COMT2 is expressed constitutively or from early stage in pif4 and g-box mutants. Is it the case? This will strengthen authors' claim of the central importance of SI-phyB1/SI-PIF4/G-box at proSI-COMT2.

Response: Thank you! Previous studies indicated that during tomato fruit ripening, DNA methylation is the key regulatory component (Zhong et al.,

2013; Lü et al., 2018). We checked the DNA methylation rate of *SICOMT2* in our unpublished tomato genome methylation Database. We found that at the green stage, the *proCOMT2* was highly methylated. Therefore, even without the inhibition of SIPIF4, the expression of *SICOMT2* is low during green fruit stages in *slpif4* or *proslcomt2* mutants. Actually, this is the key advantage of our gene-editing for *proCOMT2*, we only removed the SIPIF4 inhibition during the ripening stages without changing its expression pattern in other stages. We have add the DNA methylation data of *SICOMT2* as Fig. S28. We also added this discussion to lines 393-400.

Fig. S28 The DNA methylation rate of *SICOMT2* at the green stages (30DPA, 40DPA) and ripening stages (49DPA, 55DPA). DPA (days after anthesis).

References

Zhong, S., Fei, Z., Chen, Y., et al., Single-base resolution methylomes of tomato fruit development reveal epigenome modifications associated with ripening. *Nature Biotechnology*, 2013, 31 (2): 2462.

Lü, P., Yu, S., Zhu, N., et al., Genome encode analyses reveal the basis of convergent evolution of fleshy fruit ripening. *Nature Plants*, 2018, 4: 784-791.

7. All methods should be written in detail. For example, authors wrote that they injected MG132 (10 mM MgCl₂, 50 µM MG132) 6 h before the collection. But how much? Please go through the method section and describe various protocols more carefully. Below are more or less editorial points.

Response: Thank you! A clear description has been presented in the method (Lines 569-580): “80mM MG132 (10mM MgCl₂, 50µM MG132) and its reference solution were injected 6h before collection. A CCD camera was used to observe luciferase activity in tobacco leaves. For the western blot, 30 DPA tomato fruits were selected from RNAi-SlphyB2 and wild-type plants, and inject infection solution from the bottom of the fruit until there is liquid leaching at the stem. Incubate the infected fruits in the dark for 24 hours, followed by 3 days of dark cultivation. Half of the plants with the injected MG132 or its reference solution were treated with red light for 30 minutes. BCA was used for the determination of the total protein concentration. SDS-PAGE electrophoresis was performed

with the consistent protein content of each sample.”

8. Please include at least a representative original metabolic profile for each in a supplementary figure. Currently, authors provided only C/MS running profiles for standard chemicals, thus, it is difficult to judge authors' claims.

Response: Thank you! We have provided the original mass spectra (second order spectrum) in supplemental Fig. S4 and extracted ion chromatogram (XIC) in supplemental Fig. S6, S12, S15, S16, S20, S26, S27.

Fig. S4 The mass spectra (MS/MS spectrum) of melatonin and its intermediates detected by LC/MS.

Fig. S6 The extracted ion chromatogram (XIC) of melatonin and its intermediates detected by LC/MS in instantaneously transformed samples. (associated with Fig. 1c).

XIC of melatonin

Fig. S12 The extracted ion chromatogram (XIC) of melatonin detected by LC/MS in stable transgenic tomato. (associated with Fig. 2a).

Fig. S15 The extracted ion chromatogram (XIC) of melatonin detected by LC/MS in wild-type and transgenic tomato fruit under control light (CL) and red light supplement (RLS). (associated with Fig. 3c).

Fig. S16 The extracted ion chromatogram (XIC) of N-acetylserotonin and 5-methoxytryptamine detected by LC/MS in wild-type tomato fruit under control light (CL) and red light supplement (RLS). (associated with Fig. 3b).

Fig S20 The extracted ion chromatogram (XIC) of melatonin detected by LC/MS in wild-type and RNAi-*SIP4* transgenic tomato fruit under control light (CL) and red light supplement (RLS). (associated with Fig. 4i).

Fig. S26 The extracted ion chromatogram (XIC) of melatonin detected by LC/MS in wild-type and RNAi-*SlphyB2* transgenic tomato fruit under control light (CL) and red light supplement (RLS). (associated with Fig. 5f).

Fig S27 The extracted ion chromatogram (XIC) of melatonin detected by LC/MS in wild-type and T2 CRISPR (CR) tomato fruit. (associated with Fig. 7).

9. Authors wrote (n=3) for metabolite analysis and RT-qPCR analysis. I presume they are from three different tomato plants. Please indicate clearly what n=3 means in figure legends or method sections.

We have added detailed descriptions of sample collection and statistics in the corresponding place of the methods and legends section. Unless specifically mentioned, three biological replicates were calculated for most experiments. For T₀ transgenic plants, each individual fruit from the

same seedlings is recognized as one biological replicate. Please see the figure legends of Fig S11, S19 and S25. For WT and T1 plants, one biological replicate is the pool of 10-12 fruit from the same seedling. Please see the figure legends of Fig 1c, 2a, 3a-c, 4d, 4g-i, 5e-f, 7c, S7 and S14.

10. Authors are using 'melatonin synthase' genes indicate melatonin biosynthetic genes. Since synthase refers a specific type of enzyme, I suggest not to use synthase in this context.

Response: Thank you! Modifications have been made.

11. Conventionally, non-italic capital lettered PHYB refers apoprotein, while phyB refers a holoprotein. Since authors are likely dealing with a holoprotein, please use phyB instead of PHYB.

Response: Thank you! We replaced PHYB with phyB throughout this article.

12. Please indicate the line numbers in Fig S7 to relate them with S8. Same for Fig S12, S14 etc.

Response: Thank you! The line numbers have been added in the related figures. Please see new Fig. S10, S11, S19 and S25.

13. Fig S11 is incomprehensible. Please label each lane properly and

write down detailed figure legends.

Response: Thank you! We have labeled each lane in Fig. S11 and added detailed figure legend. Please see New Fig. S18.

14. All suppl figures supporting the main figure should be cited in main figure legends and vice versa. For example, Fig S15 seems to be associated with Fig 5d. Please cite each other in figure legends. An arrow in Fig S15 is misleading: the arrow indicates as if a band in SDS gel is PIF4. Authors may rather include the full immunoblot image of Fig 5d in suppl Fig S15. I'm not quite sure the utility of SDS PAGE of plant cell extracts. Please also see #4.

Response: Thank you! We mentioned “to which main figure this supplementary figure is associated with” in most of supplementary Figure legend. The figure legend of Fig. S15 have been revised more accurately. And the full immunoblot image of the original Fig. 5d has been added in current Fig. S24.

Fig. S24 The detection by western blot of total protein in WT and transgenic plants of RNAi-*SlphyB2*. (a) SDS-PAGE detects the total

protein loading. The black arrow marks the position of the SIPIF4 protein as calculated according to the size of protein. (b) Full immunoblot image of western-Blot detecting for degradation of SIPIF4 protein. R1 and R2 are two technical replicates. This supplementary data related to the fig. 5d.

REVIEWERS' COMMENTS

Reviewer #1 (Remarks to the Author):

I am sorry, but still the methods, after one round of revision, are not complete, and thus they are not adequate to support the results: in my opinion this work cannot be published.

Reviewer #2 (Remarks to the Author):

All of my concerns on the previous manuscript were dealt properly. I have three minor editorial comments.

1. SlphyB2 vs SIPHYB2 (*italic*): please differentiate protein vs gene. For phytochrome nomenclature, please refer the following article (<https://www.ncbi.nlm.nih.gov/pmc/articles/PMC160449/pdf/060468.pdf>)

2. In the revised method section, authors wrote "80mM MG132 (10mM MgCl₂, 50μM MG132) and its reference solution were injected". Is 80mM a typo of 80mL? However, 80mL seems to be awfully a lot to be injected to a single MicroTom fruit. Please make sure the number and unit are accurate not only for this but throughout the revised manuscript.

3. 'Synthase' is still used in Fig # legend.

REVIEWERS' COMMENTS

Reviewer #1 (Remarks to the Author):

I am sorry, but still the methods, after one round of revision, are not complete, and thus they are not adequate to support the results: in my opinion this work cannot be published.

Response: More detailed methods had also been added in the Materials and Methods section (Lines 494-500, 509-510, 615-627).

Reviewer #2 (Remarks to the Author):

All of my concerns on the previous manuscript were dealt properly. I have three minor editorial comments.

1. SlphyB2 vs SIPHYB2 (italic): please differentiate protein vs gene. For phytochrome nomenclature, please refer the following article (<https://www.ncbi.nlm.nih.gov/pmc/articles/PMC160449/pdf/060468.pdf>)

Modifications have been made according to the article.

2. In the revised method section, authors wrote "80mM MG132 (10mM MgCl₂, 50μM MG132) and its reference solution were injected". Is 80mM a typo of 80mL? However, 80mL seems to be awfully a lot to be injected to a single MicroTom fruit. Please make sure the number and unit are accurate not only for this but throughout the revised manuscript.

The description have been corrected. It is "1mL 80 μM MG132 (10 mM MgCl₂, 80 mM MG132)".(Line 577-578)

3. 'Synthase' is still used in Fig # legend.

Modifications have been made.